# Residents' Willingness to Pay for a Carbon Tax

**Ie Zheng Goh** [1] and **Nitanan Koshy Matthew** [1,2,*]

1. Department of Environment, Faculty of Forestry and Environment, Universiti Putra Malaysia, Serdang 43400 UPM, Selangor, Malaysia; nickey.goh@gmail.com
2. Institute of Tropical Agriculture and Food Security (ITAFoS), Universiti Putra Malaysia, Serdang 43400 UPM, Selangor, Malaysia
* Correspondence: nitanankoshy@upm.edu.my

**Abstract:** Addressing environmental issues has been a significant challenge. Malaysia is one of the fastest-growing countries in terms of economic, social, and land use development but high in $CO_2$ emission rates. The introduction of a carbon tax is seen to reduce greenhouse gases emission (GHG), but the uncertain extent of implementation, based on economic theory, remains unknown. Hence, the current study's objectives are to assess residents' knowledge and attitude towards GHG. It is also to analyse the factors influencing residents' Willingness to Pay (WTP). Three hundred and eleven (311) residents from Klang were selected using convenience sampling. The result shows that most of the respondents were willing to pay and had medium knowledge and a high level of attitude towards GHG. Poisson regression analysis results showed that gender, age, income, education, number of households, and marital status variables significantly influenced the maximum WTP. Overall, the residents' WTP for a carbon tax was estimated at RM36.31 per year for open-ended (CVM): RM36.96 and double bound (CVM): RM35.65. A mechanism such as investment in green technology, eco-transportation, and green energy using the tax can be applied. This study is pivotal towards achieving SDG 13: Climate action.

**Keywords:** greenhouse gases emissions; willingness to pay; carbon tax; knowledge; attitude; contingent valuation method

## 1. Introduction

### 1.1. Introduction and Problem Statement

Addressing environmental issues has been a significant challenge for many countries striving for sustainable economic development [1]. Carbon dioxide is identified as one of the main components of greenhouse gases and can be used to produce diesel fuel [2,3]. These days, the use of fossil fuels has become a worldwide issue. Human activity in development has released large quantities of carbon dioxide ($CO_2$) and other greenhouse gas into the atmosphere [4].

Human activities, including burning fossil fuels, deforestation, land development, and electricity production contribute to climate change. In the end, this climate change will cause a greenhouse effect. Atmospheric greenhouse gases concentrations, such as $CO_2$, methane ($CH_4$), and nitrous oxide ($N_2O$) have been attributed to human activity since the 1750's [5,6]. Emissions of greenhouse gases have contributed significantly to air pollution and affected climate changes by increasing the atmosphere's temperature [7,8]. The release of $CO_2$ gas, resulting from fossil fuels while producing fuel worldwide, has also contributed to global warming [1].

The most significant increase in energy consumption and $CO_2$ emissions occurred in cities, especially where growing populations enjoyed higher living standards and material prosperity [9]. Increasing demand for energy resources also affects living standards through urbanisation and industrialisation [10]. This is because the increase in demand for energy resources, especially fossil fuels, will increase $CO_2$ emissions. As a response, about

40 national jurisdictions and more than 20 cities, states, and regions have implemented or planned an explicit carbon price, covering approximately $7GtCO_2e$, accounting for about 12% of global annual greenhouse gas emissions [8,11]. The number of carbon pricing tools implemented or planned has increased from US$20 to US$38 [12]. There has been concern that carbon pricing will damage industrial competitiveness. As such, most clear prices are still low, about less than US$10 per ton of carbon dioxide only, and there is no mechanism or plan to increase them [11]. Some countries also provided exemptions or special treatment for their most polluting energy-intensive industries, thereby limiting the effectiveness of the carbon price [11]. For example, the British Government pledged to reduce greenhouse gas emissions by at least 80% by 2050 [13]. In order to reduce the emissions, they introduced the Contracts for Difference (CFD), Renewable Obligation (RO), and Feed in Tariff (FIT) exemption schemes to all the industries [13].

In Southeast Asia, Malaysia is one of the fastest-growing countries in terms of economic, social, and land use development [14]. The $CO_2$ emission rate in Malaysia in 2018 measured 2210.6 ton, while in 2017 it was only 2123.3 ton. In 2016, the $CO_2$ emission rate measured 2044.1 ton [15]. Many sources of $CO_2$ have led to increased emissions in Malaysia [16]. Coal power plants function as one of Malaysia's major sources of $CO_2$ emissions [17]. Expanding tourism development will also increase $CO_2$ emissions in Malaysia [18]. This is because tourist arrivals in Malaysia will increase $CO_2$ emissions through transportation services [19]. Industrialisation in Malaysia can also create pollution in the environment, resulting in rising $CO_2$ emissions [20].

Malaysia's $CO_2$ emissions are mainly caused by electricity consumption, mobility, and municipal solid waste accumulated in landfills [21]. $CO_2$ emissions in Malaysia are associated with fossil fuels for the production of commodities and from the household sector's demand [22]. Not only that, but particulate matter 10 ($PM_{10}$) also exceeds the Malaysian air quality guideline in Petaling Jaya, Gombak, Kelang, Kajang and Kuala Lumpur. This is affecting human health already [23]. The primary contributor sources of $PM_{10}$ in Malaysia include power generation, motor vehicles, and industries [24,25]. Usually, the elderly, children, patients with respiratory problems, heart disease, and allergy patients are the victims of the effects of particulate matter [26,27]. When air pollution rises to a dangerous level, the fatality rate will peak [28]. In order to raise standards and enact pollution control measures, local and national governments increasingly gather cost and benefit information about the level of pollution levels to support them in overcoming the issue [23]. Hence, a WTP study is helpful to estimate the economic benefits of air pollution reduction.

Economic valuation is defined as a measurement of the economic value of the benefit of conservation. It reveals a price for ecosystem services to provide information to decision-makers, and hence, facilitates quantification of the trade involved and help in the decision-making process [29]. Willingness to Pay (WTP) measures the maximum amount of money an individual is willing to pay to increase the quality of an item or service that can be experienced [30].

Few countries' citizens also faced issues about citizen acknowledgement about air pollution. In China, most of the public had little knowledge level concerning the impacts of air pollution on natural resources availability and people's health [31]. Attitudes towards air pollution is low, especially towards climate change and WTP for reducing air pollution [32]. In Germany, consumers lacked knowledge and information about air pollution and the voluntary carbon offset market [33]. The citizens there also experienced low or zero knowledge of voluntary carbon offsetting. Knowledge about carbon offset will influence the demand for voluntary carbon offsets [34]. The resident's or citizen's attitudes may lead to good intentions. However, it must be noted that certain elements such as social norms, lack of knowledge, change of behaviour, and education may act as a barrier in combating air pollution [35].

In Malaysia, many citizens did not acknowledge aspects of air pollution. Many Malaysians did not have any experience or ideas about carbon reduction programmes

in combating air pollution [36]. Some Malaysians also did not know about the origin and source of air pollution [37]. They also lacked sufficient knowledge of environmental protection and conservation [38]. The typical citizen attitude towards air pollution is somewhat thoughtless. Only a few Malaysians have experience buying a carbon offset initiative [36]. Most of them were not interested in paying for air quality improvement because they lack environmental awareness [39]. Malaysian citizens also have a negative attitude towards public transportation as they preferred to drive their private vehicles [39]. Overall, knowledge about and attitudes towards air pollution will affect the WTP, which is particularly important [40]. The citizen's understanding of air pollution may influence their attitudes towards the effectiveness of those policies set by The Intergovernmental Panel on Climate Change. The citizen's practices and lifestyle could influence the emission of greenhouse gases [41].

There are scant studies done in Malaysia on the WTP for a carbon tax that involves the residents' greenhouse gas emissions reduction. This is because a large majority of studies focused on the developed country only. No study or research evidence has investigated the issue in Malaysia, or developing countries, on a carbon tax.

Therefore, Klang, Selangor was chosen as the research area due to the air pollution index of Malaysia. Klang holds the highest air pollution index among Kuala Selangor, Petaling Jaya, Shah Alam, and Banting [42]. The statistic by the Department of Statistics Malaysia stated that, throughout the year 2018, Klang holds the highest record in August, which is 227API. The overall result in Selangor is the status index of air quality in Klang, where 271 days is good, 91 days is moderate, and two days is unhealthy [43]. The cargo and container traffic in Port Klang, also known as the busiest port in Malaysia, has many imports and exports, and contributes to this situation [44].

The choice of the area was considered strategic because there are many residents in Klang. As the country grows, the population will also increase. In 2018, there were 1,025,000 people in Klang, of which 552,400 were male, and 472,700 were female, compared to 2017 and 2016, where the population had a total of 1,008,000 and 991,600 individuals, respectively [43]. The annual population growth rate in Klang was 1.6% in 2018, 1.7% in 2017, and 1.8% in 2016 [43]. However, of 1,025,000 individuals, 117,000 are non-citizens [43]. The rise in population will increase the emissions rate due to cascade effects serving people's needs [7]. Pollution, such as fossil fuel use, increases the number of buildings and cars, will cause a rise in greenhouse gas emissions [45].

Hence, this study focuses on estimating the WTP for a carbon tax in Malaysia using the double bound CVM and the open-ended CVM method for comparison purposes. The specific objective of the study is to assess Klang residents' knowledge and level of attitude towards greenhouse gases emission. It is also to analyse the factors influencing the residents' willingness to pay for a carbon tax generally and estimate their willingness to pay for a carbon tax in Klang specifically. It will guide future consideration in determining the carbon tax and hopefully help achieve sustainable development for future generations. Not only that, but this research will also help to achieve the Sustainable Development Goals (SDG) in Malaysia, principally SDG 13: Climate Action, which is mainly focused on reducing GHG emissions [8].

### 1.2. Literature Gap

Existing studies as shown in Table 1 include general studies on WTP for air quality improvements through greenhouse gases emissions reduction [31,32,39,45–51]. Some studies focused on specific aspects of transportation with a focus on vehicle owners, such as those by Brouwer, Brander, and Van Beukering [52]; Gupta [53]; Rotaris and Danielis [54]; Rizali et al. [30]; Schwirplies et al. [34] and airline services by Jou and Chen [55]; Shaari et al. [36]. Other aspects include companies' carbon emissions trading schemes [56]. Hence, in terms of a literature gap, following Zhang; Wang; Sun and Liu [57], only 12% of the total world publications on carbon tax were from 1991 to 2014. In addition, general studies found on WTP for greenhouse gas emissions reduction are slightly outdated and not getting much

attention in ASEAN countries per se, including Malaysia. Therefore, there is a need to conduct a study that focuses on the residents' willingness to pay for a carbon tax to ensure the knowledge on this subject matter is updated from time to time.

**Table 1.** Summary of existing literature on carbon tax.

| Author | Country | Type of Carbon Tax | Method of Study | Factors | WTP (Price per Unit) |
|---|---|---|---|---|---|
| Brouwer et al. (2008) | United Kingdom | Emission based | CVM (Open ended, Double bounded) | Nationality, Flying frequency, Awareness, Price ticket, Household income | €25 (RM122.02) per ton $CO_2$-eq |
| Carlsson et al. (2010) | Sweden, China, United States | Emission based | CVM (Open ended, Payment card) | Gender, Age, Household size, Education, Income, Religious, Political affiliation | 2000 SEK (RM972.88) per year per household |
| Diederich and Goeschl (2011) | Germany | Emission based | CVM (Single bounded, Payment card) | Cash prize, Gender, Age, Number of children, Education, Personal benefit, Future benefit, Lifestyle, Carbon footprint | €6.30 (RM30.75) per ton of $CO_2$ |
| Tsang and Burge (2011) | United Kingdom | Emission based | CVM (Iterative bidding) | Level of income, Social-economic background | Between £1.45 (RM7.97) and £2.97 (RM16.33) per year |
| Blasch (2013) | Switzerland | Emission based | CVM (Single bounded, Payment card) | Age, Gender, Academic level, Monthly gross income, Marital status, Knowledge of offsetting | 78 CHF (RM349.35) per $tCO_2$ |
| Duan et al. (2014) | China | Emission based | CVM (Open ended, Iterative bidding) | Gender, Annual income, Education, Political orientation, Member of environmental organisation, House ownership, Car ownership | CNY201.86 (RM124.29) per year or CNY16.82 (RM10.36) per month for each person |
| Jou and Chen (2015) | Taiwan | Emission based | CVM (Open ended, Single bounded) | Education level, Annual number of flights, Monthly income, Age, Gender | NT$39.05 (RM5.91) per passenger |
| Tolunay and Başsüllü (2015) | Turkey | Emission based | CVM (Open ended, Payment card) | Unplanned urbanisation, Residence, Age, Gender, Marital status, Occupation, Number of household members, Income per capita | US$23.52 (RM94.61) per consumer |
| Gupta (2016) | India | Emission based | CVM (Open ended, Single bounded) | Interest, Environmental activeness, Use of public transport, Quality of public transport, Age, Education, Family size, Individual income | Rs581.5 (RM32) per people |
| Bazrbachi et al. (2017) | Malaysia | Emission based | CVM (Single bounded) | Gender, Age, Efficiency of public transport, Education level, Health index, Income, Air pollution concern | RM4.99 per trip |
| Akhtar et al. (2017) | Pakistan | Emission based | CVM (Open ended, Single bounded) | Gender, Age, Education level, Marital status, Number of children, Number of households, Monthly income, Air quality area | US$9.86 (RM39.66) per month or US$118 (RM474.65) per year |
| Jones et al. (2017) | United States | Emission based | CVM (Open ended, Single bounded) | Age, Education, Gender, Ideology, Income, Attitudinal belief | US$3.66 (RM14.72) per year per household |
| Kotchen et al. (2017) | United States | Emission based | CVM (Single bounded, Double bounded) | Education, Gender, Household size, Income, Age | US$177 (RM711.98) annually |
| Rizali et al. (2017) | Indonesia | Emission based | Open ended CVM | Car ownership, Level of education, Car insurance availability | Rp 432.182,70 (RM1225.12) average per year |
| Schwirplies et al. (2017) | Germany | Emission based | CVM (Iterative bidding) | Level of contribution, Politics, Religious, Age, Gender, Number of children, Education level, Residents, Compensation scheme | €52 (RM253.80) or €53 (RM258.68) per $tCO_2$e |

**Table 1.** *Cont.*

| Author | Country | Type of Carbon Tax | Method of Study | Factors | WTP (Price per Unit) |
|---|---|---|---|---|---|
| Nastis and Mattas (2018) | Greece | Emission based | CVM (Open ended, Single bounded) | Age, Education level, Level of income, Household size, Gender | €81 (RM395.34) per household |
| Zhao et al. (2018) | China | Emission based | CVM (Open ended, Double bounded) | Types of company, Carbon market, Potential, Sector type, Company size, Experience | 35 yuan (RM22.81) per $tCO_2e$ |
| Rotaris and Danielis (2019) | Italy | Emission based | CVM (Single bounded, Double bounded) | Attitudes and belief, Environmental awareness, Political affiliation, Place of residents, Car ownership, Gender, Age, Education, Employment status, Income level | €101 (RM492.95) to €154 (RM751.63) per litre |
| Shaari et al. (2020) | Malaysia | Emission based | CVM (Open ended, Double bounded) | Bid price, Income, Gender, Age, Education, Job, Offset information, Occupation | RM86.00 per passengers |

## 2. Materials and Methods

### 2.1. Research Location

The study was conducted in Klang, Selangor, as shown in Figure 1. Klang is part of Klang Valley, which is the primary economic zone in Malaysia [58]. The land area of Klang is 632 km$^2$ [59]. There are nine districts in Klang Valley which include Gombak, Hulu Langat, Hulu Selangor, Klang, Kuala Langat, Kuala Selangor, Petaling, Sabak Bernam and Sepang [60]. The state legislative area consists of Kota Anggerik, Batu Tiga, Kota Kemuning, Sungai Kandis, Sentosa, Pandamaran, Bandar Baru Klang, Pelabuhan Klang, Selat Klang, Sementa and Meru.

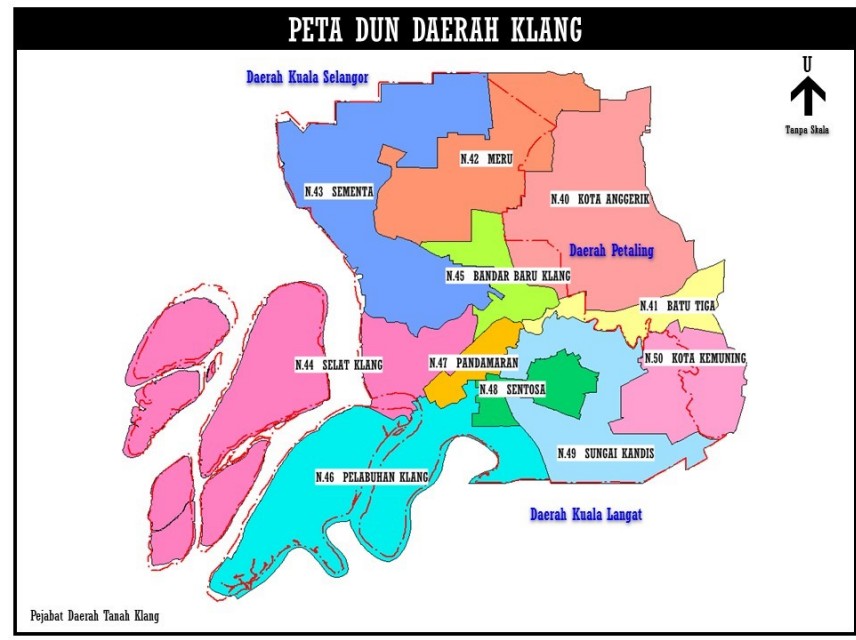

**Figure 1.** Geographical map of Klang, Selangor. Source: Klang Land and District Office [60].

### 2.2. Model Specification for Double Bound CVM and Open-Ended CVM

Table 2 shows the variables used in the analysis of WTP.

Double Bound CVM

$$WTP = \beta0 + \beta1Price + \beta2Environment\ attitude + \beta3Gender + \beta4Age +$$
$$\beta5\ Monthly\ gross\ income + \beta6Education + \beta7Household\ size +$$
$$\beta8Marriage\ status + \mathcal{E}$$

Open-Ended CVM

$$MAX\ WTP = \beta0 + \beta1Environment\ attitude + \beta2Gender + \beta3Age +$$
$$\beta4\ Monthly\ gross\ income + \beta5Education + \beta6Household\ size +$$
$$\beta7Marriage\ status + \mathcal{E}$$

**Table 2.** Variables-variables used in the analysis are listed below.

| | **Dependent Variable with 1 If a Respondent Is Willing to Pay for the Amount Asked to Them, 0 Otherwise** |
|---|---|
| | Maximum (MAX) WTP |
| Initial BID | Bid price levels set out in the CVM question (Dichotomous choice format) RM 5, RM 10, RM 15, RM 20 |
| BID2 | Follow-up the bid assigned |
| Environment attitude | Likert scale |
| Gender | 1 for male, 2 for female |
| Age | Age of the respondent (years) |
| Monthly gross income | Income of the respondents (RM/month) |
| Education | 1 for primary, 2 for secondary, 3 for diploma, 4, degree, 5 for master/PhD, 6 for others |
| Family size | Household size of the respondents (people) |
| Marital status | 1 for single, 2 for married |
| $\mathcal{E}$ | Random error |

*2.3. Research Design*

2.3.1. Data Sources

This study uses primary data and secondary data. For secondary data involved the use of reading sources: libraries, texts, journals, magazines, and reports to assist in collecting data and information to facilitate the process of completing research. For example, the researcher obtained additional information from the Klang Town Planning Department that is unavailable in the written material in libraries or published on a website.

For all the objectives, primary data were collected. This research uses quantitative methods to obtain data. This method only involved the use of questionnaires. The questionnaire was distributed to the targeted respondents. Researchers distributed the questionnaires online using an online platform to the targeted respondents living in the vicinity of Klang, Selangor. Internet surveying is expected to obtain higher response rates and is more affordable than face to face or phone based surveying [61]. Online platforms such as Facebook, Messenger and WhatsApp were used to distribute the questionnaire. Besides that, the chain referral technique was used, where friends and family members who reside at Klang can fill and subsequently refer to their friends, colleagues, and neighbours. Email services were also used to email the questionnaire to the government and non-governmental offices to boost the response rate. Before that, a letter of application of distribution was sent to the authorities to approve distribution to make sure proper permissions were obtained and the flow of data collection was smooth. Finally, with all parties' cooperation, all the data were collected and analysed, supported, and presented.

2.3.2. Questionnaire Design and Structure

The questionnaires were provided in English. The questionnaires consist of closed and open-ended questions.

The questionnaire has five main sections. Section one covers the air quality in Klang, Selangor. Section two is the residents' knowledge of greenhouse gases emission. Section three involved the residents' attitude towards greenhouse gases reduction. Section four covers their willingness to pay. Section five includes respondent demographics.

In the questionnaire, both the Open-ended and Single Bound and Double Bound CVM elicitation techniques were used for the willingness to pay part. For the latter technique, the dichotomous choice format question was used. A double-bounded logit model is more efficient than the single-bounded as the value obtained is deemed to be more reliable about the respondent's willingness to pay [62]. For Double Bound CVM: four different sets were used in which the WTP price is different in terms of the starting bid price for CVM question which set A (RM5), set B (RM10), set C (RM15), and set D (RM20). Based on Bid 1: RM5 (Set A), if the option "yes" is chosen, the WTP amount will be increased by RM5 for each bid. If the option "no" is chosen, then the bid's lower amount will be presented as shown in Table 3. Next, for open-ended CVM, the respondents were asked about the maximum value they are willing to pay for air pollution as shown in Table 3. The respondents were given a scenario about air pollution as shown below:

**Table 3.** Willingness to Pay (WTP).

| |
|---|
| **D1.Let us say a household must pay RM 5 a year for a carbon tax in Klang, Selangor. Are you willing to pay?**<br>☐ **Yes (please proceed to question D2)**<br>☐ **No (please proceed to question D3)** |
| **D2**. Let us say a household must pay **RM 10 a year** for a carbon tax in Klang, Selangor. Are you willing to pay?<br>☐ **Yes (please proceed to question D4)**<br>☐ **No (please proceed to question D4)** |
| **D3**. Let us say a household must pay **RM 2.50 a year** for a carbon tax in Klang, Selangor. Are you willing to pay?<br>☐ **Yes (please proceed to question D4)**<br>☐ **No (please proceed to question D4)** |
| **D4**. What is the maximum amount that you are willing to pay for a carbon tax Klang, Selangor? **(please state)**<br>● **Maximum payment is** RM per year. |

Example of scenario:

Klang is well known for being the most polluted city in Selangor state, with the highest index of Air Pollution Index (an air quality measurement) almost every day. Klang also has the third-highest population in Selangor. The increase in population in Klang throughout the years also will result in further impacts on the environment, especially air pollution. With the money collected through a carbon tax, activities such as investment in new sustainable energy, technology (e.g., solar energy, wind energy, energy-efficient cars), awareness program, policy, and many more will be proposed to help in greenhouse gases reduction.

This study will help understand residents' willingness to pay for a carbon tax in Klang, Selangor.

Before answering the following question, think about:

● The amount of willingness to pay is based on the ability to pay once every year.

Based on the scenario above, please mark (√) how much you are willing to pay for a carbon tax in Klang, Selangor.

### 2.4. Validity and Reliability Analysis

Before distributing the questionnaire to respondents, the questionnaire was validated by five experts in the field.

Validity is an instrument in which an idea is precisely measured in quantitative research [63]. Validity is also an essential term in instrument development [64]. Validity only calculates what it wants to measure. There are three types of validity: criterion validity, content validity, and construct validity [63,64]. For this study, only content validity was used to minimise the potential error for the questionnaire.

Content validity refers to the extent to which the study instrument measures all aspects of the structure accurately [63]. Content validity can help improve the possibility of obtaining the effectiveness of the support structure at a later stage [65].

The researchers selected a total of five panels of experts in economics to measure the instrument. Each panel received a questionnaire to provide their comments and evaluate them. The personnel from the panels conducted a four-point scale questionnaire to rate them by. The point scale used was 1 = not relevant, 2 = somewhat relevant, 3 = relevant, 4 = very relevant. All these scales were used in all sections of the questionnaire for each question. Content Validity Index (CVI) and Aiken's V method was conducted to analyse the point filled by the panel to obtain the accurate result. However, Scale-level-CVI (S-CVI) was calculated using the number of items in the tool that received a "very relevant" rating [66]. It is recommended that scales with excellent content validity be 0.78 or higher for I-CVI while for S-CVI/UA and S-CVI/Ave for 0.8 and 0.9 or higher, respectively [67]. In addition, ideally, a value of 1.00 in I-CVI should be present if there are five or fewer judges, and in the case of six or more judges, I-CVI should not be less than 0.78 [65]. Furthermore, another researcher supported that an S-CVI/Ave value of more than 0.9 has excellent content validity [66]. It is recommended that a minimum S-CVI should be 0.8 for reflecting content validity [65]. Otherwise, the question will have to change or be removed to gain better validity.

Fifty items were identified in the questionnaire. The result shows the average validity for all sections using Aiken's formula gained 0.73 for all sections. However, S1 gained 0.74, followed by S2, S4, and S5, each gaining 0.73. Lastly, S3 gained 0.68 only. Overall, the result shows that content validity is slightly lower than the proposed 0.78 [65]. Therefore, an adjustment towards the questionnaire was made.

After validation, comments, and suggestions were shared, the researcher made the necessary correction. After correction, the final questionnaire was distributed to the respondents.

Next, for reliability, a pilot study was conducted with 40 respondents. For Section 2: knowledge, the reliability was Kuder-Richardson coefficient of reliability (K-R20). The result shows that the value obtained is 0.6347, which ranked strong (0.61–0.79), as shown in Table 4.

**Table 4.** Kuder-Richardson coefficient of reliability rank.

| Reliability Coefficient | Level of Reliability |
| --- | --- |
| 0.81 or more | Near complete agreement |
| 0.61–0.80 | Strong |
| 0.41–0.60 | Moderate |
| 0.21–0.40 | Fair |
| 0.00–0.20 | Poor agreement |

Source: Kuder and Richardson [68].

Next, for section three (Residents' attitude towards greenhouse gas emissions), Cronbach's Alpha strength analysis tested the reliability. The result showed that the Cronbach's Alpha values obtained is 0.804 for only 12 questions in total, whereby the level of reliability is good. Therefore, after removing two questions, question number five and eight, the results of Cronbach's Alpha values turn out to be 0.860, suggesting that the level of reliability is very good, as shown in Table 5.

**Table 5.** Cronbach's Alpha.

| Reliability Coefficient | Level of Reliability |
| --- | --- |
| 0.90 or more | Very good |
| 0.80–0.89 | Good |
| 0.60–0.79 | Normal |
| 0.40–0.59 | Doubted |
| 0.00–0.39 | Rejected |

Source: Faizal, Lee, Leow, Wei; Pallant [69,70].

### 2.5. Open-Ended CVM

Mean WTP can be easily affected by the assumed formation of the end of the distributions [71]. Mean WTP will be measured by confirming no negative amount from the respondents for a carbon tax using the equation proposed by Honu [72] as shown in Equation (1).

$$Mean\ WTP = \frac{1}{N}\sum_{i}^{N} WTP_i \tag{1}$$

Source: Honu [72].

### 2.6. Sampling Technique

The total population in 2018 in Klang, Selangor was 1,025,000 people, and 552,400 were male while 472,700 were female [15]. Thus, after computing using the formula by Yamane [73], as shown in Equation (2), the total number of respondents needed for this research was 400. The study respondents were selected using non-probability sampling, which is an easy sampling method to select respondents. The sampling procedure was taken using convenience sampling, where the respondent was easier to reach. The lack of a sample size frame is suitable for this type of sampling [74].

$$n = \frac{N}{1 + N(e)^2} \tag{2}$$

Source: Yamane [73].

$$n = \frac{1025000}{1 + 1025000(.05)^2}$$
$$= 399.844 \tag{3}$$
$$\approx 400$$

Nonetheless, the study could not obtain the required sample size, which was possible for 311 only. Although the study did not meet the requirement as proposed by Yamane [73], using the G* Power 3.1 application for sample size calculation, the minimum sample size required for a multiple regression analysis such as Poisson regression was estimated at only 89 samples based on (Effect size: $f^2$) = 0.15, $\alpha$ err prob: 0.05, Power (1-$\beta$ err prob) = 0.95 and (6) number of predictors. Therefore, 311 respondents were sufficient for analytic purposes.

### 2.7. Data Analysis

The analysis used in this study was descriptive analysis and inferential statistical analysis. This research used percentage statistics, mean scores, standard deviations, frequency, crosstab, central tendency distribution, and standard deviation for descriptive analysis. The data were retrieved and analysed using descriptive analysis of mean scores using the Statistical Package for Social Sciences (SPSS) software version 25 to obtain the frequency and mean scores and then draw a conclusion to obtain the results. The mean score value was determined based on Lendal's [75] guidance, which interprets the mean score according to the mean score average set. The overall result was used to carry out the answer for objectives one and two. For inferential statistical analysis, this research used Poisson regression for open-ended CVM and logit regression for double bound CVM. The data was prepared and analysed using the STATA software version 15 to identify

the coefficient and p-values by looking at the variables. Descriptive analysis was used to answer objective one for knowledge and attitude, covering percentage statistics, frequency, and mean scores. However, inferential statistics analysis was used to answer objectives two and three, involving a regression model. Both analyses were used to answer the objectives and research questions.

## 3. Results and Discussion

### 3.1. General Information on Respondent Demographic

Table 6 shows the respondents' demographic data. The result shows that female respondents were 57.6% (179) of the respondents, while male respondents were 42.4% (132). Female respondents are more sensitive and worry about health and environmental problems [36]. This is different from the result from Chang [41]; Diederich and Goeschl [51], whereby male respondents were more common than females, as females said they were not familiar with such a statement and are unwilling to participate in the survey. As for the age factor, the result shows that the highest category is age 18–25 years, which was 35.4% (110) of the respondents. However, the lowest category is the age higher than 56 years at 8% (25) of the respondents. The elderly are unwilling to pay for air quality improvement as it will not benefit them [23]. Age is also one of the variables determining the willingness to pay [36]. The result also shows that 79.1% (246) of the respondents have a degree in education. However, only 0.3% (1) of the respondents choose other qualifications in their qualification level. The higher the respondents' level of education, the higher the willingness to pay for carbon offset [36]. The result also shows that marital status shows few differences, in which marital status was 50.5% (157) of the respondents, and a single status was 49.5% (154) of the respondents. The marital status does not influence the willingness to pay to improve air quality [39].

**Table 6.** Respondent demographic.

| Gender | Frequency (*n* = 311) | Percent |
|---|---|---|
| Male | 132 | 42.4 |
| Female | 179 | 57.6 |
| **Age** | | |
| 18–25 | 110 | 35.4 |
| 26–35 | 84 | 27 |
| 36–45 | 56 | 18 |
| 46–55 | 36 | 11.6 |
| >56 | 25 | 8 |
| **Education level** | | |
| Primary | 5 | 1.6 |
| Secondary | 7 | 2.3 |
| Diploma | 14 | 4.5 |
| Degree | 246 | 79.1 |
| Master and PhD | 38 | 12.2 |
| Others | 1 | 0.3 |
| **Marital status** | | |
| Single | 154 | 49.5 |
| Married | 157 | 50.5 |
| **Number of individuals within a household (including you)** | | |
| 1–3 people | 59 | 19 |
| 4–6 people | 217 | 69.8 |
| >7 people | 35 | 11.2 |

**Table 6.** *Cont.*

| Gender | Frequency (*n* = 311) | Percent |
|---|---|---|
| **Employment status** | | |
| Student | 76 | 24.4 |
| Self-employed | 44 | 14.1 |
| Government sector | 69 | 22.2 |
| Private sector | 95 | 30.5 |
| Retired | 22 | 7.1 |
| Others | 5 | 1.6 |
| **Monthly gross household income (overall)** | | |
| B40 (<RM4360) | 190 | 61.1 |
| M40 (RM4361–RM9619) | 75 | 24.1 |
| T20 (>RM9619) | 46 | 14.8 |
| **Monthly gross income** | | |
| <RM2000 | 92 | 29.6 |
| RM2001–RM3000 | 59 | 19 |
| RM3001–RM4000 | 46 | 14.8 |
| RM4001–RM5000 | 31 | 10 |
| >RM5001 | 83 | 26.7 |

Results also show that the number of households from four to six people has the highest percentage, which was 69.8% (217) of the respondents. In comparison, the number of households with more than seven people have the least respondents, at only 11.2% (35) of the respondents. The increasing number of adults will decrease the willingness to pay for improving air quality [39]. Approximately 30.5% (95) of respondents work in the private sector, while only 1.6% (5) work in the others sector. Most of the respondents who work in the professional sector are willing to pay more than the non-professional sector [23,36]. There are 61.1% (190) of the respondents whose monthly gross income is in the B40 category, while there were 14.8% (46) of the respondents whose monthly gross income is in the T20 category. A respondent with a higher income is willing to pay more since they can afford it even at a higher price [36]. People in a high-income category, have an illness, or are able to witness the depletion of air quality are more likely pay to improve the air quality [39]. As much as 29.6% (92) of the respondents earn a monthly gross income less than RM2000, while only 10% (31) of the respondents earn a monthly gross income from RM4001 to RM5000. This is supported by Fong et al. [7] that the higher the income, the higher the energy used, and emission produced.

*3.2. General Information on Air Quality in Klang, Selangor*

Table 7 shows a total of 311 respondents from the distribution of the questionnaire. Regarding satisfaction with air quality in Klang, results show that 69.5% (216) of the overall respondents were not satisfied with the air quality. In comparison, 30.5% (95) of the respondents were satisfied with the air quality. This is because Klang is currently undergoing an urbanisation process, which leads to an increase in population. The increase in population will eventually lead to the occurrence of many pollutions, including air pollution. This is supported by Fong et al.; Tolunay and Başsüllü and Chen [7,45,76] who argued that the rapid increase in population will increase the emissions rate.

**Table 7.** Air quality in Klang, Selangor.

| Are you satisfied with the air quality in Klang? | Frequency (*n* = 311) | Percent |
|---|---|---|
| Yes | 95 | 30.5 |
| No | 216 | 69.5 |
| **Are you concerned about the air pollution in the community where you live?** | | |
| Yes | 253 | 81.4 |
| No | 58 | 18.6 |
| **How severe would you say is the air pollution in the community where you live?** | | |
| Low | 49 | 15.8 |
| Moderate | 230 | 74 |
| High | 32 | 10.3 |
| **How would you feel about the quality of air pollution?** | | |
| Worried | 246 | 79.1 |
| Not worried | 65 | 20.9 |
| **Who do you think should be primarily responsible for the reduction of air pollution?** | | |
| Government | 11 | 3.5 |
| Citizen | 13 | 4.2 |
| Industries | 41 | 13.2 |
| Non-Governmental Organisation | 9 | 2.9 |
| All the above | 237 | 76.2 |
| **What is your most favourite way to obtain knowledge related to air pollution and related protective measures?** | | |
| Television | 42 | 13.5 |
| Internet | 227 | 73 |
| Books | 7 | 2.3 |
| Newspaper | 9 | 2.9 |
| Lecturer | 4 | 1.3 |
| Friends | 21 | 6.8 |
| Others | 1 | 0.3 |
| **Are you aware of the greenhouse gases emission reduction measure?** | | |
| Yes | 236 | 75.9 |
| No | 75 | 24.1 |
| **If you were responsible for designing a plan to address greenhouse gases emission reduction, which of the following technologies would you use? (Multiple responses possible)** | | |
| Solar energy | 189 | 13.5 |
| Energy-efficient appliances | 120 | 73 |
| Energy-efficient cars | 123 | 2.3 |
| Wind energy | 82 | 2.9 |
| Nuclear energy | 26 | 1.3 |
| Carbon capture and storage | 53 | 6.8 |

Note: For the last questions, the respondent may choose more than one answer.

For the concern about air pollution in the respondents' community, results show that 81.4% (253) of respondents were concerned about the air pollution in the community they lived, while only 18.6% (58) respondents were not concerned. This is because air pollution can lead to various illnesses and diseases. This is supported by Gupta [53] that the increase in local air pollution can become a dilemma in health and welfare impacts.

About 74% (230) of the respondents rank their air pollution level in their community to be moderate. This is because the areas in which they reside are not exposed to anthropogenic activities. This is supported by Chang [41] that most of the respondents answered that man-made activities cause pollution. However, 15.8% (49) of respondents ranked their air pollution level in the community as low. This is because their community area might be considerably far from the industrial or town area. Air pollution can contribute to many adverse side effects, especially in human health, agriculture, and industrial production [39]. Only 10.3% (32) of respondents ranked their air pollution level in the community they live in as high. This is because they live near urban areas exposed to pollution, as the city's heart tends to cause air pollution resulting from more anthropogenic activities. This is supported by Rotaris and Danielis [54] that urban communities are more vulnerable to air pollution resulting from transportation and others.

There were 79.1% (246) of respondents worried about air pollution quality, while only 20.9% (65) were not worried about the quality of air pollution. Exposure to significant air pollution can lead to many dangerous diseases that contribute to various health problems. Thus, respondents are very aware of it. This is supported by Krupnick, Rowe, Lang; Cropper, Simon, Alberini, Arora, Sharma [77,78] who argued that the rise in air pollution levels will lift the public concern rate. Individuals that live in highly polluted areas are willing to pay more than the slightly polluted area [23].

Taking responsibility for the reduction in air pollution is essential in combating air pollution. A total of 76.2% (237) of respondents selected "all of the above" that everyone (government, citizen, industries, non-governmental organisation) should be primarily responsible for reducing air pollution. This is because air pollution can be solved with the cooperation of all parties. However, only 2.9% (9) of the respondents selected that non-governmental organisation should be responsible for reducing air pollution. This is because non-governmental organisations such as The Clean Air Forum Society of Malaysia (MYCAS) can help share more information effectively throughout the whole nation. Companies and the authorities should work together to control air pollution [36]. This is supported by Chang [41] that the central and local governments, industrial firms, non-governmental organisations, international organisations, and individuals and families should be responsible for air pollution. This is also supported by Fong et al. [7] that researchers and policymakers oversee reducing air pollution. Research by Akhtar et al. [39] shows that 65.5% of the respondents believe that every citizen should be responsible for pollution.

A total of 73% (227) respondents prefer to obtain knowledge related to air pollution and related protective measures by the Internet; nowadays, getting information online is much more efficient and affordable. Internet surveys tend to obtain higher response rates than face to face and phone surveys [61]. This is supported by Chang [41] that 70% of his respondents obtain awareness through mass media. However, only 0.3% (1) of respondents obtained such knowledge and measure through family members.

For awareness of greenhouse gas emission reduction measures, 75.9% (236) of respondents are aware of it, while only 24.1% (75) respondents are unaware of it. This is because the air pollution issue is prevalent globally. Thus, it became a hot topic in news coverage. This is supported by Ameyaw and Yao [79] that global environmental change is crucial for humans. Among those technologies provided to address greenhouse gas emission reductions, those 236 respondents choose solar energy to be their highest priority at 31.87% (189). This is because solar energy is clean and renewable. The majority of the respondents support the ideology that solar energy is one of the ways to reduce carbon from the atmosphere [61].

Secondly, an energy-efficient car was an option chosen by a total of 20.74% (123) of respondents. This is because energy-efficient cars primarily use electrical energy, which produces less pollution and is better for the environment than fossil fuel-based cars. Eco-friendly transportations can reduce air pollution significantly [7,80]. This is supported by Chang [41] that his respondents can help reduce pollution by bringing down daily transportation. Thirdly, energy-efficient appliances are chosen by 20.23% (120) of respon-

dents. This is because energy-efficient appliances use lesser energy to operate. Thus, it can help reduce electricity bills. This is also supported by Chang [41] who showed that his respondents were willing to pay for green products. Wind energy was selected by 13.82% (82) of respondents supporting it. This is because wind energy can produce electricity that can minimise the use of burning fossil fuels. The local support for wind energy is highly supported as it will decrease the annual electricity costs [81]. Subsequently, carbon capture and storage only accumulate 8.93% (53) of respondent choice. This is because this strategy is too expensive in terms of capturing it. Curry [61] mentioned that a majority of the respondents did not know about carbon capture and storage before. Lastly, nuclear energy has the lowest percentage, with only 4.38% (26) of the respondents. This is because nuclear energy is too dangerous if it is not managed correctly and professionally. Therefore, it is firmly rejected by the respondents. This is supported by Curry [61] that the public is puzzled about using nuclear power plants to solve climate change issues.

### 3.3. General Information on Residents' Knowledge of Greenhouse Gases Emission

Table 8 shows the residents' knowledge of greenhouse gases emissions. From the result, we can identify that 98.4% (304) of respondents know that global warming is one of the issues most countries face, while only 0.3% (1) of the respondents are unsure about this. Fong et al. [7] also mentioned that global warming is one of the issues most countries face. The majority of the respondents acknowledged global warming and its consequence of increasing temperature [61]. This is also supported by Shah et al. [1] that the release of carbon dioxide gases contributed to global warming worldwide. Next, respondents also know that the costs for carbon sequestration service can ensure that future generations live healthily when most of the respondents, or 69.1% (215), answered yes. This is different from the research by Curry [61] that carbon sequestration terms are unfamiliar to the respondents. Therefore, most of them are unsure about it. However, only 5.5% (17) of respondents do not know about it. In addition, the result shows that 61.4% (191) of the respondents know that political changes will affect climate regulation, while only 13.8% (43) of the respondents are unsure about this statement. This is supported by Rotaris and Danielis [54] that political understanding is one factor affecting environmental policy. Besides that, 37.3% (116) of the respondents know about carbon offset, while 29.3% (91) of the respondents are unsure about carbon offset. This is not supported by Blasch [47] who showed only 17% of the respondents are aware of carbon offset. The same goes for research by Shaari et al. [36] that almost half of the respondents have no idea about carbon offset, and only a few got involved in it.

Furthermore, the result shows that 83.6% (260) of the respondents do not know that temperature has not increased globally. However, only 8% (25) of the respondents were unsure about this statement. This is different from research done by Harris Interactive [82], whereby 74% of the respondents believe that carbon dioxide and other harmful gases can contribute to global warming and finally lead to an increase in temperature. Plus, 83.9% (261) of the respondents know that the emission of the carbon monoxide causes air pollution, while only 4.5% (14) of the respondents are unsure of such a statement. Emission of carbon monoxide from transportation, industrial and power plants released into the atmosphere will result in air pollution [61].

Moreover, the result shows that 93.2% (290) of the respondents know that the emission of waste gases causes air pollution. However, only 1.3% (4) of the respondents do not know about that. Furthermore, 88.4% (275) of the respondents know that investing in energy-saving technology can aid in combating air pollution, while only 3.2% (10) of the respondents do not know about this statement.

**Table 8.** Residents' knowledge of greenhouse gases emission.

| Item | Frequency (n = 311) | | | Ranking |
|---|---|---|---|---|
| | **Yes** | **No** | **Do Not Know** | |
| Knowledge 1 | 306 (98.4) | 4 (1.3) | 1 (0.3) | 1 |
| Knowledge 2 | 215 (69.1) | 17 (5.5) | 79 (25.4) | 6 |
| Knowledge 3 | 191 (61.4) | 77 (24.8) | 43 (13.8) | 12 |
| Knowledge 4 | 116 (37.3) | 104 (33.4) | 91 (29.3) | 8 |
| Knowledge 5 | 26 (8.4) | 260 (83.6) | 25 (8) | 10 |
| Knowledge 6 | 261 (83.9) | 36 (11.6) | 14 (4.5) | 5 |
| Knowledge 7 | 290 (93.2) | 4 (1.3) | 17 (5.5) | 4 |
| Knowledge 8 | 275 (88.4) | 10 (3.2) | 26 (8.4) | 11 |
| Knowledge 9 | 290 (93.2) | 16 (5.1) | 5 (1.6) | 2 |
| Knowledge 10 | 86 (27.7) | 209 (67.2) | 16 (5.1) | 7 |
| Knowledge 11 | 32 (10.3) | 263 (84.6) | 16 (5.1) | 9 |
| Knowledge 12 | 286 (92) | 18 (5.8) | 7 (2.3) | 3 |

Knowledge 1: Global warming is one of the issues faced by most countries. Knowledge 2: Costs for carbon sequestration service can ensure that future generations live in a healthy manner. Knowledge 3: Political changes will affect climate regulation. Knowledge 4: I know about carbon offset. Knowledge 5: The temperature has not increased globally. Knowledge 6: Emission of carbon dioxide causes air pollution. Knowledge 7: Emission of waste gases causes air pollution. Knowledge 8: Investing in energy-saving technology can help in combating air pollution. Knowledge 9: Global climate change is already taking place. Knowledge 10: Global climate change is not happening now, but it will happen in the future. Knowledge 11: Global climate change will not occur at all. Knowledge 12: Humans have caused the temperature to increase. Note: The ranking ranges from 1 to 10, signifying participants' knowledge from most to least knowledgeable.

Results also show that 93.2% (290) of the respondents know that global climate change is already taking place. This is supported by research done by Rotaris and Danielis [54] that most of their samples believe that climate change has occurred. On the other hand, only 1.67% (5) of the respondents do not know about this statement. Besides that, we can identify that 67.2% (209) of the total respondents do not know that global climate change is not happening now. However, considering if it will happen in the foreseeable future, only 5.1% (16) of respondents do not know about this statement. In addition, the result shows that 84.6% (263) of the respondents do not know that global climate change will not occur at all, while only 5.1% (16) of the respondents are unsure about this statement. Lastly, humans have caused the temperature increase; the result shows that 92% (286) of the respondents know about it, while only 2.3% (7) of the respondents are not sure about this statement.

*3.4. General Information on Residents' Attitude towards Greenhouse Gases Emission Reduction*

Table 9 show residents' attitude towards greenhouse gas emission reduction. First, it clearly shows that every effort towards climate protection is effective because almost all the respondents strongly agree with this statement and obtain the highest percentage, which is 36.7% (114) of respondents. In comparison, only 1.9% (6) of the respondents strongly disagreed with this statement. Next, we can also notice that the residents would also contribute part of their income if they were sure that the money would be used to prevent atmospheric pollution with 36.7% (114) of the respondents agreeing with it. However, there are only 4.5% (14) of the respondent who strongly disagreed with the statement. Respondents would pay more if they were convinced that quick action would prevent pollution [61]. Besides that, for statements educating younger generations about environmental protection is important shows that 72.7% (226) of the respondents strongly agree with it (e.g., encourage carpool). However, only 0.3% (1) of the respondents disagreed with this statement. Promoting environmental education should be implemented during primary school age [36].

**Table 9.** Shows residents' attitude towards greenhouse gases emission reduction.

| Item | Frequency | | | | | Mean | Ranking | Score Level |
|---|---|---|---|---|---|---|---|---|
| | 1 | 2 | 3 | 4 | 5 | | | |
| Attitude 1 | 6 (1.9) | 16 (5.1) | 74 (23.8) | 101 (32.5) | 114 (36.7) | 3.9678 | 5 | High |
| Attitude 2 | 14 (4.5) | 23 (7.4) | 91 (29.3) | 114 (36.7) | 69 (22.2) | 3.6463 | 7 | Medium |
| Attitude 3 | 1 (0.3) | 0 (0) | 8 (2.6) | 76 (24.4) | 226 (72.7) | 4.6913 | 1 | High |
| Attitude 4 | 2 (0.6) | 9 (2.9) | 77 (24.8) | 80 (25.7) | 143 (46) | 4.1350 | 4 | High |
| Attitude 5 | 25 (8) | 43 (13.8) | 128 (41.2) | 66 (21.2) | 49 (15.8) | 3.2283 | 10 | Medium |
| Attitude 6 | 9 (2.9) | 43 (13.8) | 112 (36) | 78 (25.1) | 69 (22.2) | 3.4984 | 9 | Medium |
| Attitude 7 | 11 (3.5) | 14 (4.5) | 89 (28.6) | 108 (34.7) | 89 (28.6) | 3.8039 | 6 | High |
| Attitude 8 | 6 (1.9) | 51 (16.4) | 90 (28.9) | 87 (28) | 77 (24.8) | 3.5723 | 8 | Medium |
| Attitude 9 | 0 (0) | 3 (1) | 44 (14.1) | 112 (36) | 152 (48.9) | 4.3280 | 2 | High |
| Attitude 10 | 3 (1) | 3 (1) | 44 (14.1) | 131 (42.1) | 130 (41.8) | 4.2283 | 3 | High |

Attitude 1: Every single effort towards climate protection is effective. Attitude 2: I would contribute part of my income if I were certain that the money would be used to prevent atmospheric pollution. Attitude 3: Educating younger generations about the knowledge of environmental protection (ex. encourage carpool) is important. Attitude 4: Reduction in the use of air-conditioning can be made by me to improve the current atmospheric situation. Attitude 5: Protecting the environment should be given priority, even it might increase the unemployment rate. Attitude 6: I often cut back on driving a car to protect the environment. Attitude 7: Protecting the environment is necessary, even if it will slow down economic growth. Attitude 8: Government must bear the full cost of reducing air pollution. Attitude 9: Citizen is responsible for climate change. Attitude 10: I feel obligated to protect the climate.

Moreover, the result shows that there are 46% (143) of the respondents who know that reducing of the use of air-conditioners can be made with an improvement of the current atmospheric situation, while only 0.6% (2) of the respondents strongly disagreed with this statement. Reducing pollution is a more efficient way than energy conservation to prevent climate change [61]. Furthermore, we can also identify that 41.2% (128) of the respondents feel neutral about the statement that protecting the environment should be given priority even though it might increase the unemployment rate. However, only 8% (25) of the respondents strongly disagreed with this statement. This is supported by Fong et al. [7] who showed it is essential to regulate environmental quality, although it may reduce job opportunity, education, and quality of life. Enhancing the environment quality can offer more job opportunities [83].

Plus, we can know that 36% (112) of the respondents feel neutral about cutting back often on driving a car to protect the environment. Meanwhile, there are only 2.9% (9) of the respondents who strongly disagreed with this statement. This result is supported by Fong et al. [7] that reducing the transportation sector can protect the environment. In addition, the result shows that 34.7% (108) of the respondents agreed that protecting the environment is necessary, even it will slow economic growth, while only 3.5% (11) of the respondents strongly disagreed with that. Air pollution can result in economic loss. Therefore, the public put pressure on the authorities to approach the situation with action [39].

On the other hand, the result shows that 28.9% (90) of the respondents feel neutral about the government's statement that the government must bear the total cost of reducing air pollution. In comparison, only 1.9% (6) of the respondents strongly disagreed with this statement. Respondents believe that they should not bear the cost of reducing air pollution, but the authorities and companies of service should instead [36]. Authorities must put effort into controlling pollution [39].

Furthermore, the result shows that 48.9% (152) of the respondents strongly agreed that the citizens are responsible for climate change, whereas only 1% (3) of the respondents disagreed with such a statement. Ideally, passengers and the public must be well educated about climate change [36]. Lastly, 42.1% (131) of the respondents agreed that they feel obligated to protect the climate. However, only 1% (3) of the respondents strongly disagreed

with this statement that they feel obligated to protect the climate. This is supported by Akhtar et al. [39] that most respondents worry about the air quality as it is crucial towards their health.

### 3.5. General Information on the Willingness to Pay Responses

#### 3.5.1. Single Bound CVM and Double Bound CVM

The variables used from the single bound CVM and double bound CVM are gender, age, number of households, income, education, marital status, knowledge, attitude, and gross income to identify the *p*-value. However, the result shows that all the variables are insignificant, whereby the WTP estimations using the bid 1 and bid 2 variables is acceptable. Hence, to solve this situation, this study was conducted using the Poisson regression for open-ended CVM, where the dependent variables are the maximum WTP. Poisson regression can allow no random item in the variables, and the variance does not contain an error component [84]. Therefore, most of the p-value results show significance at level 1% and 10%.

Table 10 shows the Poisson regression residents' maximum WTP as the dependent variable. All the results below are based on the collection of 311 respondents from the questionnaire. The result shows that the variable gender is significant at a 1% level. A negative coefficient for gender means that the male is more willing to pay more than the female. This shows that male respondents are more willing to pay for environmental improvement. This result is inconsistent with Safian and Hamzah; Shaari et al. [36,80] that females are more willing to pay for environmental conservation. The result shows that variable age is also significant at the 1% level. A negative coefficient for age means that the younger respondents are more willing to pay for a carbon tax. This is supported by Safian and Hamzah; Lu and Shon [40,80] that younger respondents believed in paying for environmental protection. However, this is different from the research done by Shaari et al. [36] that the older respondents are more willing to pay for a carbon tax to ensure the future generations' environment is conserved. The result shows that variable income is also significant at the 1% level. A positive coefficient for income indicates that the higher the monthly gross income of the respondents, the more they are willing to pay for a carbon tax. This is supported by Fong et al. [7] that a passenger with high earning is willing to pay for carbon prevention to minimise emissions and pollution.

**Table 10.** Poisson regression.

| Variables | Coef. | Std. Err. | z | $p > |z|$ | [95% Conf. Interval] | |
|---|---|---|---|---|---|---|
| Gender | −0.0991372 | 0.0191298 | −5.18 | 0.000 *** | −0.136631 | −0.0616434 |
| Age | −0.007413 | 0.0009105 | −8.14 | 0.000 *** | −0.0091975 | −0.0056285 |
| Income | 0.0000177 | $2.30 \times 10^{-6}$ | 7.68 | 0.000 *** | 0.0000132 | 0.0000222 |
| Education | 0.1040986 | 0.0155919 | 6.68 | 0.000 *** | 0.0735391 | 0.1346581 |
| Number of households | −0.046041 | 0.0060985 | −7.55 | 0.000 *** | −0.0579939 | −0.0340881 |
| Attitude | −0.0141071 | 0.0199381 | −0.71 | 0.479 | −0.0531851 | 0.0249709 |
| Marital status | −0.0376537 | 0.0213806 | −1.76 | 0.078 * | −0.0795588 | 0.0042514 |
| _cons | 3.840645 | 0.1167101 | 32.91 | 0.000 *** | 3.611898 | 4.069393 |

Note: *** significant at 1% level of confidence; * Significant at 10% level of confidence.

The result shows that variable education is significant at a 1% level. A positive coefficient for education means that the higher the level of education obtained by the respondents, the more the willingness to pay for a carbon tax. This is because educated respondents know more aspects concerning environmental issues [36]. This is supported by Masud, Al-Amin, Akhtar, Kari, Afroz, Rahman MS, Rahman [85] that education level plays a role in determining the willingness to pay a respondent for a carbon tax. The result shows that a variable number of households is significant at the 1% level. A negative coefficient for the number of households means that when the number of households is lower, the more

willing they are to pay a carbon tax. This is because it will cause a burden to a big family, especially low-income families. Akhtar et al. [39] proved that the increasing number of family members decreases the willingness to pay for improved air quality. The result also shows that the attitude of the respondents is not significant. The result shows that the marital status of the respondents is significant at the 10% level. A negative coefficient for marital status means that if the respondents are singles, they are more willing to pay a carbon tax. Tolunay and Başsüllü [45] proved that the singles would be willing to pay for air quality improvement than those who are married, divorced, or widowed.

### 3.5.2. Estimation of WTP for Open-Ended and Double Bound CVM

The result shows that the mean WTP (based on the open-ended CVM) for this research is RM36.97 per year, while double bound CVM is RM35.65 per year. Comparing with the double bound WTP result, this defines that the amount of WTP is higher by RM1.32 only. The proposed average value is RM36.31 per year. Parts of Malaysians based in Klang, Selangor, are willing to pay more money and contribute part of their income for environmental protection [86].

The mean WTP findings by Safian and Hamzah [80] show that Malaysian consumers are willing to pay about RM6.5 yearly towards environment protection, which is lower than our result. In addition, the mean WTP findings worth a total of US\$3.66 (RM14.82) over 20 years are willing to pay by the respondents in Colorado River, United States, to reduce greenhouse gas emissions [50]. Compared to our study, the result is lower than our mean WTP. However, the mean WTP findings from a study by Shaari et al. [36] mentioned that passengers in Malaysia are willing to pay RM86 per trip for a carbon offset, which is higher than our result to reduce the emissions. Lastly, the mean WTP findings from Tolunay and Başsüllü [45] study is higher with a maximum of US\$23.52 (RM95.26), with a willingness to pay by the community in Turkey to implement carbon sequestration services.

Overall, the result is acceptable. Therefore, the average maximum WTP for a carbon tax is RM36.31 per year.

### 3.5.3. Open-Ended CVM

The mean WTP can be calculated by dividing the sum of $WTP$, starting with $WTP_i$ and ending with $WTP_N$, with the total number of items of $WTP$ where 1 is the total amount of $WTP$, $N$ is the total number of respondents, $i$ is the exponential function.

$$Mean\ WTP = \tfrac{1}{311}(11497)$$
$$= 36.97$$

### 3.5.4. Double Bound CVM

Table 11 shows the double bound CVM. Maximum WTP is the dependent variable in this study. The result shows the maximum willingness to pay by the respondents towards carbon tax based on double-bound contingent valuation methods worth RM35.65 only. The government can levy as much as this amount per year if the carbon tax proposal is proposed.

**Table 11.** Double bound CVM.

|  | Coef. | Std. Err. | z | $p > |z|$ | [95% Conf. Interval] | |
|---|---|---|---|---|---|---|
| Beta_cons | 35.64819 | 3.2316 | 11.03 | 0.000 *** | 29.31437 | 41.98201 |
| Sigma_cons | 21.98887 | 3.072689 | 7.16 | 0.000 *** | 15.96651 | 28.01123 |

Note: *** significant at 1% level of confidence.

### 3.5.5. Reasons

Figure 2 shows the reasons why people are willing to pay. There are many reasons why people are willing to pay a carbon tax. Based on this research, the result shows that the main reason that people are willing to pay for a carbon tax is that they feel responsible for their

contribution to climate change, which shows a total of 37%. This is supported by Shaari et al. [36] that a traveller is presumably willing to pay extra for their airfare to minimise emissions. The second reason that influences them is that they care about the environment in general, which accumulate 32.5%. The younger generations are more responsible for the environment [36]. The third reason, which obtained 14.1% of the respondents, was to avoid future natural disasters. People wish to preserve the forest to ensure that future generations are safe from future disasters [45,51]. The fourth reason, which has a percentage of 7.4% from the respondents, states that the environment has the right to be protected irrespective of the costs. The rest of the reasons, to reduce future economic damage costs and not willing to share their opinion, each obtained 6.8% and 2.3% from the respondents.

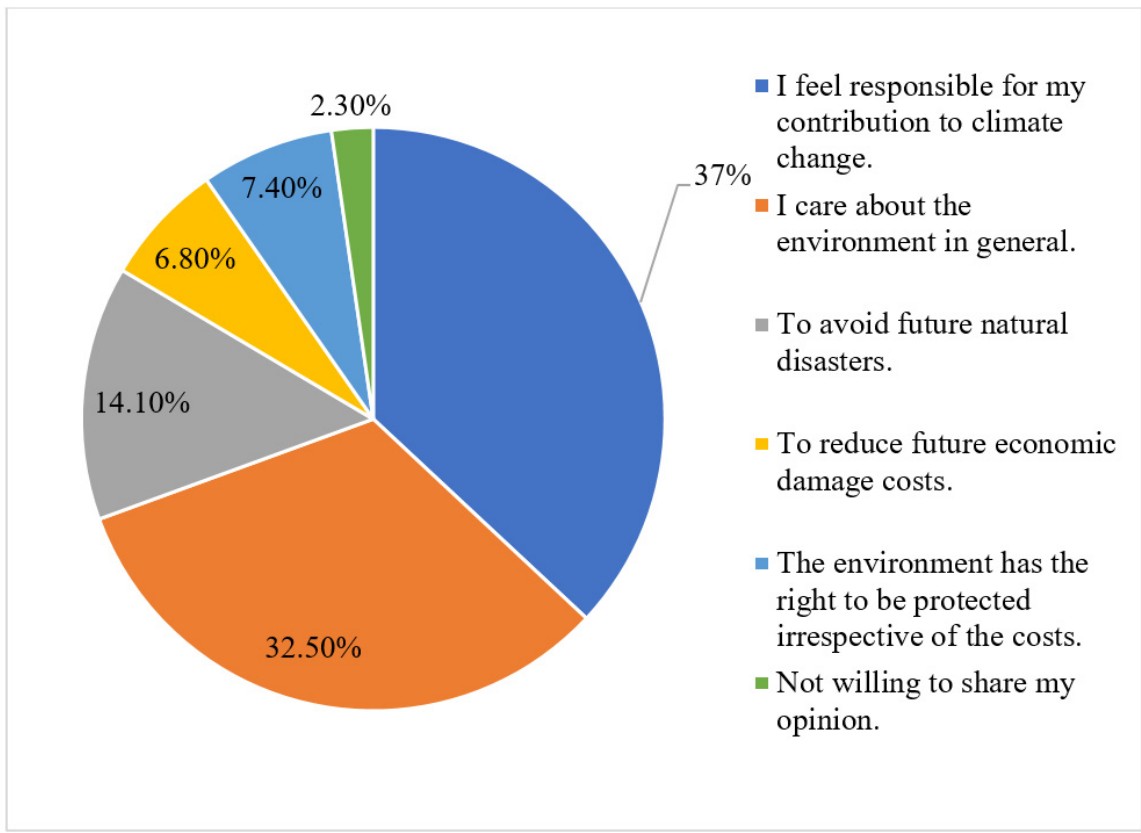

**Figure 2.** Reasons why people are willing to pay.

Figure 3 shows some reasons why people are not willing to pay. There are many reasons why people are not willing to pay a carbon tax. Based on this research, the result shows two main reasons why people are not willing to pay for a carbon tax, which each accumulates a total of 33.8%. First is their income being too low. This is supported by Chang [41] that the low-income level would become a burden for the respondents, especially when the prices are too high. The second is that they do not believe that such a program would have any real impact. This is supported by Rotaris and Danielis [54] that the failure of government programmes on environmental motive has reduced the public's confidence towards those programmes. However, this is not supported by Blasch [47] that the respondents appreciate the authority's effort on carbon offset. The third reason which obtains a 17.4% from the respondent is that they prefer to spend their money on other things. The fourth reason is that they are unwilling to share their opinions, which shows 7.1% of the respondents. The rest of the reason is that climate change does not affect them or their family, and they feel irresponsible for their contribution to climate change which each obtain a result of 4.2% and 3.9%. Some respondents think that emissions reduction is not their responsibility [36]. Women are more willing to pay and be responsible for their

children to ensure their future [36]. This is supported by Chang [41], whereby an estimation of 30% of his respondents said that global warming is not their responsibility, and it is something they cannot control. Research by Shaari et al. [36] mentioned that respondents are also not willing to pay as airfare costs are already high.

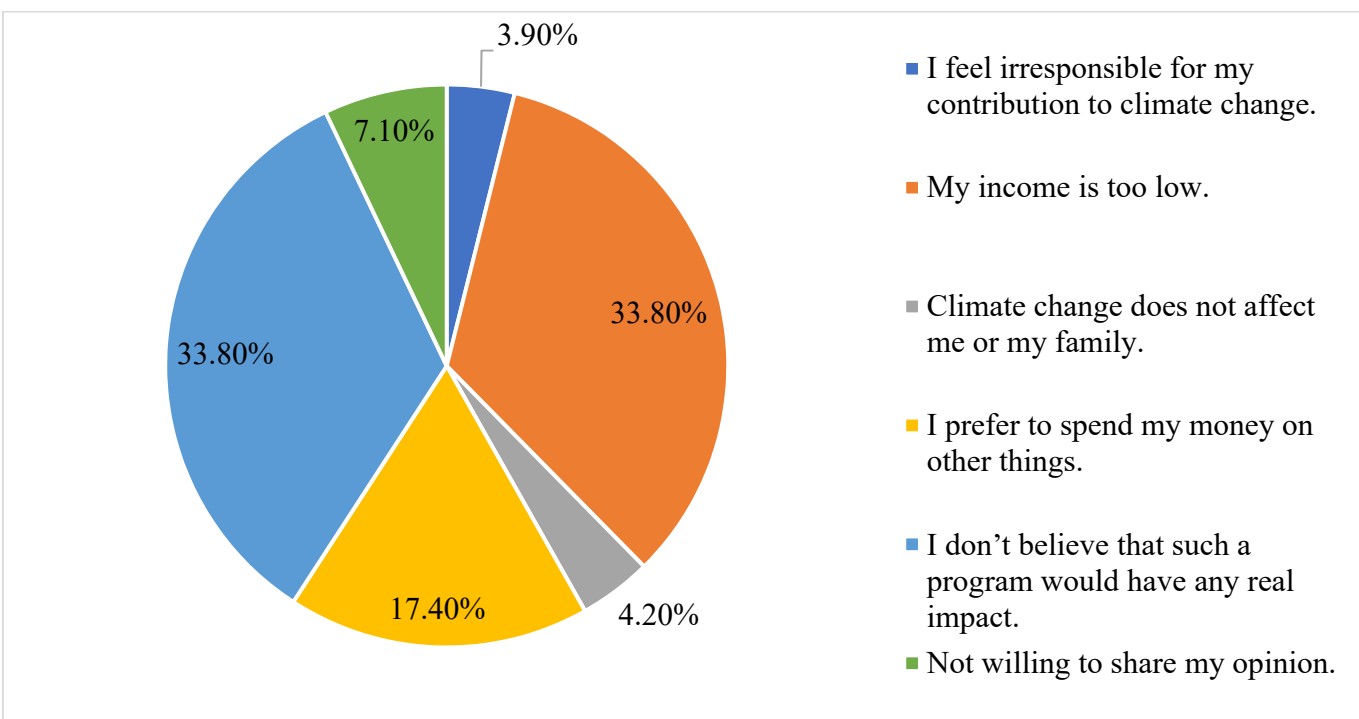

**Figure 3.** Reasons why people are not willing to pay.

Figure 4 shows the administrator deemed to be appropriate to collect the money obtained from this carbon tax. This research shows that the Ministry of Science, Technology and Innovation has the highest percentage, 61.4% of the respondents. The second administrator hat is appropriate to collect the money obtained from this carbon tax is the Inland Revenue Board Malaysia. This is the second highest which obtain a percentage of 21.2% from the respondents. Next, the result shows that 10.3% of the respondents think that the Ministry of Finance should collect the carbon tax. Followed by a total of 5.8% of the respondents think that other entities should collect the carbon tax. Lastly, the respondents believe that the Royal Malaysian Customs Department should collect it, which only obtains 1.3% of the total percentage.

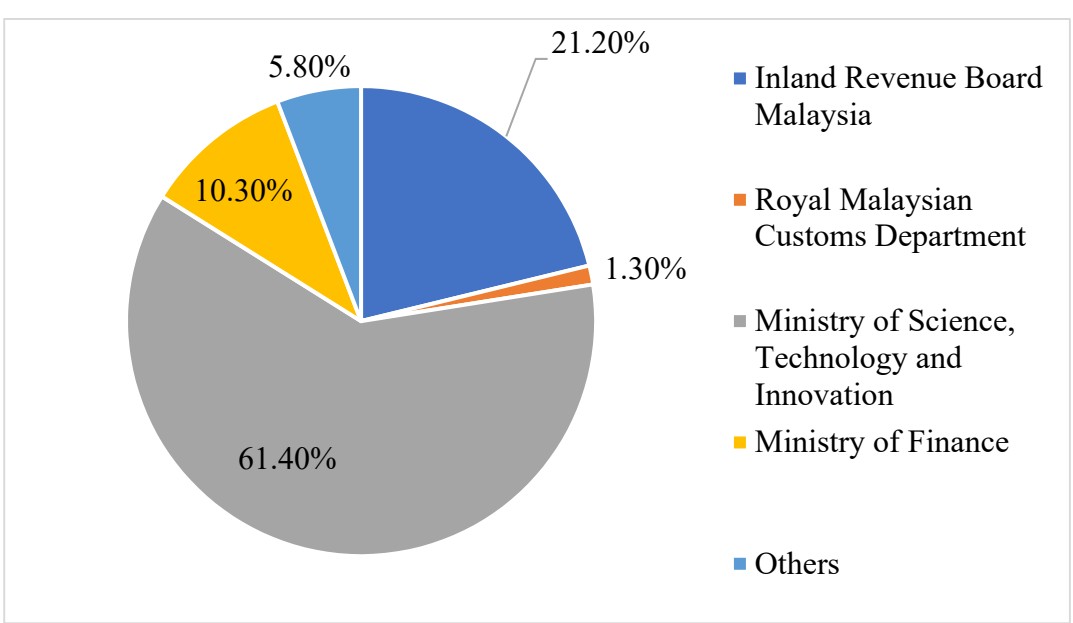

**Figure 4.** Administrator deemed appropriate to collect the money obtained from this carbon tax (ONE only).

## 4. Conclusions

This study assessed the knowledge and level of attitude towards greenhouse gases emissions and estimated the willingness to pay for a carbon tax. Most of the respondents were willing to pay because they feel responsible for their contribution to climate change. Most of the respondents are not willing to pay because their income is too low, and they do not believe that such a program would have any real impact. Most of the respondents are not satisfied with the air quality in Klang, and most of the respondents are also concerned and worried about the air pollution in their respective community areas. Moreover, respondents also rank their air pollution level in the community they live in as moderate only. Many of the respondents also think that all the above parties, including the government, citizens, industries, and non-governmental organisations, should be primarily responsible for reducing air pollution. They prefer to obtain their knowledge related to air pollution and related protective measure from the internet. Most of them are also aware of the measures concerning greenhouse gases emission reduction. Moreover, they supported the idea that solar energy is the perfect technology to address greenhouse gas emissions reduction. The result shows that the variables that significantly influenced the maximum WTP were gender, age, income, education, number of households at 1% level, and marital status at 10% level. Overall, the conclusion revealed that the average maximum of residents' WTP for a carbon tax is RM36.31 per year. The CVM was used to estimate the number of residents willing to pay for a carbon tax. The respondents also think that the Ministry of Science, Technology and Innovation should be the one who is appropriate to collect the money from this tax.

As a recommendation, the decision-makers should consider residents' preference on WTP for a carbon tax. This is to make sure that the amount is acceptable by the residents and worth the value. For example, the results derived from the residents' WTP amount from this research can be proposed as a start to the implementation process in Malaysia. Thus, it will be acceptable among Malaysians.

Other than that, the result suggests that governments can conduct awareness campaigns to expose residents to environmental issues and the importance of a carbon tax. By exposing the information about an environmental issue and the importance of carbon tax, residents will get involved in more environmental conservation activities that contribute to the environment more. For example, campaigns can be conducted using

various mediums such as newspapers, social media, broadcasts, and talks to reach a diverse audience. Thus, it will be highly informative and accessible to all parties.

Another suggestion is that the current education curriculum can also be reformed. This is to instil and expose younger generations of students or citizens about the importance of the environment. Not only that but additionally, knowledge about the current environmental issues will also be led to their acceptance of carbon taxation and its importance. In the foreseeable future, they are more willing to accept the implementation of a carbon tax. For example, reforming the education curriculum at the primary school stage can provide greater insight and awareness during early age.

Lastly, the government should serve as a role model towards citizens about carbon emission reduction. With the amount of tax collected, subsidies should be allocated to green technologies to promote clean technology. This can help build a solid and promising potential in reducing carbon emission socially and, most importantly, serve as a trial before making it mandatory. For example, the government can reduce or set green technologies to a more affordable price to encourage more citizens to invest in it to ensure a better future generation and life.

In conclusion, this research is limited in scope because it was only conducted in residents around Klang, Selangor. Therefore, it is recommended that future studies should be implemented throughout the Klang Valley or within Malaysia. Thus, all various opinions can be collected. Not only that, but it also required more respondents to gain more accurate results for this study. It is suggested that a larger number can improve the validity and reliability of the research result.

**Author Contributions:** Conceptualisation, I.Z.G. and N.K.M.; Data curation, I.Z.G. and N.K.M.; Formal analysis, I.Z.G.; Investigation, I.Z.G. and N.K.M.; Methodology, I.Z.G.; Supervision, N.K.M.; Project administration, I.Z.G.; Resources, I.Z.G. and N.K.M.; Software, N.K.M.; Validation, N.K.M.; Writing—Original draft, I.Z.G.; Writing—Review and editing, I.Z.G. and N.K.M. All authors have read and agreed to the published version of the manuscript.

**Funding:** Funding for the publication fee was provided by the Research Management Centre, Universiti Putra Malaysia, Serdang.

**Institutional Review Board Statement:** Not applicable.

**Informed Consent Statement:** Not applicable.

**Data Availability Statement:** Not applicable.

**Acknowledgments:** The authors would gratefully thank the support provided by MASPUTRA, Faculty of Forestry and Environment, Universiti Putra Malaysia. The help from the five panels of experts who validated the research instrument is remembered and appreciated. Reviews by two anonymous peers greatly enhanced the final manuscript.

**Conflicts of Interest:** The authors declare no conflict of interest.

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
