# Peer review of "Residents’ Willingness to Pay for a Carbon Tax"

_sustainability, doi:10.3390/su131810118_

Round 1
Reviewer 1 Report
Manuscript ID: sustainability-1337935
Title: Residents’ willingness to pay for carbon tax
Comments to Author(s)
General remarks:
This is a worthwhile and timely study of a significant topic. The paper provides useful information on the relationship between air pollution and residents’ willingness to pay for carbon tax among the residents in Klang, Malaysia. The paper provides relevant information, which has significant practical implication. The paper would benefit from some revision and reordering of the information presented to emphasize the main take-home messages. The paper would also benefit from additional editing and polishing to address grammatical issues, particularly the flow of sentences (there are a lot of unfinished sentences), and repetitions. I pointed out some of these errors, but there are many others. Overall, the manuscript, particularly the Introduction section is quite long, but could be condensed in several places. The observed research gap in the literature and statement of the problems need to be improved by being much more specific. Emphasis should also be given to condense Materials and Methods section. The flow would be enhanced by condensing and combining some of the contents. For example, section 2.3.2 (lines 454-456, 468-483) should be shortened in few sentences. The same occurs for sections 2.4 and 2.5 (lines 502-556), in which the two sections should be combined and shortened for a better comprehension.
Specific in-line comments can be found on the manuscript itself (attached) at places that are highlighted in blue (for editing issues) and in yellow (for those issues that require clarification and revisions). Some of the comments, however, are outlined below.
Specific suggestions for improvement
Line 8: check the font size.
Line 14: abbreviations should be written in full at first use.
Line 15: add the figure in parentheses.
Line 31: Throughout, some inconsistencies in in-text citations. In some cases, here and elsewhere, for references written by more than four authors all are mentioned while in other cases (e.g., lines 38 and 48, line 84, 109 and so on) only the name of the first author is mentioned, followed by et al.. Is this according to the Journal’s guideline?
Line 41: add “other” after the word and, and “s” on the word gas.
Line 47: use the abbreviation CO2 here and in subsequent sections of the paper.
Lines 50-110: consider revising the order of paragraphs in pages 2 and 3 or in lines 50-110. Move the text in lines 75-85 and the text in lines 95-109 after the paragraph in lines 50-60 (as seen below)
“In Southeast Asia, Malaysia is one of the fastest-growing countries in terms of economic, social and land use development (Awang, Ali, & Razman, 2019). The CO2 emission rate in Malaysia in 2018 measures at 2210.6ton while in 2017 was only 2123.3ton. In 2016, CO2 emission rate measures in 2044.1ton (Department of Statistic Malaysia, 2019). Many sources have led to increased CO2 emission in Malaysia (KNOEMA, 2020). Coal power plants act as one of the major sources of CO2 emissions in Malaysia (OECD, 2019). The expanding of tourism development will also increase CO2 emissions in Malaysia (Ling, Leong, Chuen, Vern, & Hong, 2017). This is because tourist arrivals in Malaysia will increase CO2 emissions through transportation services (Solarin, 2013). Industrialization in Malaysia can also create pollution in the environment, resulting in the rising level of CO2 84 emissions (Begum et al., 2015).
Malaysia CO2 emissions are mainly caused by electricity consumption, mobility and municipal solid waste accumulated in landfills (Khoo, 2019). CO2 emissions in Malaysia are associated with the use of fossil fuels for the production of commodities and as the demand of the household sector (Othman & Yahoo, 2014). Not only that, but particulate matter 10 (PM10) also exceeds the Malaysian air quality guideline in Petaling Jaya, Gombak, Kelang, Kajang and Kuala Lumpur and this considered affecting human health already (Masahina, Afroz, Duasa, & Mohamed, 2012). The main contributor sources of PM10 in Malaysia, including power generation, motor vehicles and industries (Department of Environment, 2020; Noor et al., 2015). Usually, the aged, children, patient with a respiratory problem, heart disease and allergy patient are the victims of the effects of the particulate matter (MOH, 2020; WHO, 2002). When air pollution rises to a dangerous level, the fatality rate will peak (Peng & Tian, 2003). In order to raise standards and taking pollution control measures, the local and national governments increasingly gather cost and benefit information about the level of the pollution levels to support them in over-coming the issue (Masahina et al., 2012).”
Then put the following paragraph (lines 86-94) after the reordered text above.
“Economic valuation is defined as a measurement of the economic value of the benefit of conservation and reveals a price for ecosystem services to provide information to decision-makers, and hence, facilitates quantification of the trade involve and help in the decision-making process (Salcone, Brander, & Seidl, 2016). Willingness to Pay (WTP) is a measure of the maximum amount of money an individual is willing to pay to obtain an increase in the quality of an item or service that can be experienced (Rizali, Sa’roni, Sopiana, & Muzdalifah, 2017). In general, the WTP is the willingness to pay the maximum price of an item to obtain goods and services (Zhao & Kling, 2004; Latumahina & Anastsia, 2014).”
Lines 65 and 67: specify if it is US$.
Lines 86-94: this paragraph seems odd here. The paragraph fits more after the text in lines 61-74 (see my suggestion above also)
Lines 92-93: repetition. Consider to remove.
Line 128: replace “did not” by “lack”
Lines 133-134: clarify Malaysian citizens also have a negative attitude towards public transportation.
Line 137: which policies? Not clear.
Lines 140-141: Repeated idea. Please remove the sentence.
Line 163: add “s”
Line 167: delete “s”
Line 167: Add "of the" after study.
Line 171: consider to replace the article "a" with "information as guidance"
Line 173: delete “SGD 13”
Line 175: the section should be shortened as there are a lot of redundancies. The authors might consider condensing the 21 paragraphs into a single paragraph or two paragraphs.
Lines 177-178: This sentence seems incomplete. Similarly, there are also incomplete sentences in lines 200-201, 211-212, 224-225 238-239 and throughout section 1.2. Please edit the errors.
Line 195: change “p” to upper case.
Line: 246: define VCO.
Line 294: make no sense. Delete.
Lines 341-342: incomplete sentence. Please revise.
Line 405: replace "uses" by "employed"
Line: 406: consider to replace “study uses” by "studies which employed"
Line 425: this variable not seen in Table 1 and elsewhere in the paper.
Table 1 unnumbered line: is that “Number of households” or “number of family members”? Better to say family size.
Lines 433-434: Delete “to obtain the information required to conduct further study”
Line 441: add “s”
Line 442: Replace “will” by "is expected to"
Line 461: please delete the sentence
Line 462: delete.
Line 464: replace “s” by “d”
Lines 502-564: please see my comment in the general remarks section above and correct these sections as per my comment.
Lines 663: Unclear. Please clarify.
Line 682: add “s”
Line 689: avoid starting a sentence with a number.
Line 694: add “s”
Lines 706-707: add “s” where suggested.
Line: 707: modify the word “family” to “families”
Line 724: add “s”
Lines 739-740: Vague, please clarify.
Table 6 unnumbered line: replace “may choose MORE THAN ONE” by “Multiple response possible”
Line 881: add “d”
Line 881: replace “is” by “are”
Line 886: delete “s”
Line 1017: add “s”

Author Response
Dear Respected Reviewer,
Thank you for your suggestion and recommendation. It is highly appreciated. Hereby I will attached the latest amendments made. Pls have a look ya. Thanks again.
Regards,
Goh

Reviewer 2 Report
Overall its a good paper. The only concern is the literature gap where there is a lack synthesis of those studies of various aspect of WTP for carbon. It good to tabulate the summary : For example: WTP (price per unit),type of carbon tax, method of study, factors, country, references).
The methodologies are well elaborated.
Results and Discussion are also well elaborated to meet the objectives of the study.
References: please check as there are some missing references such as Siti 2009
Author Response

(The authors gave the same response as above.)
